# High Expression of *CDCA7* in the Prognosis of Glioma and Its Relationship with Ferroptosis and Immunity

**DOI:** 10.3390/genes14071406

**Published:** 2023-07-06

**Authors:** Yunhan Wang, Yu Zhao, Zongying Zhang, Jie Zhang, Qiuyun Xu, Xiaorong Zhou, Liming Mao

**Affiliations:** 1Department of Immunology, School of Medicine, Nantong University, Nantong 226001, China; 2Basic Medical Research Center, School of Medicine, Nantong University, Nantong 226019, China

**Keywords:** *CDCA7*, glioma, ferroptosis, prognosis, immune

## Abstract

*CDCA7* is a copy number amplification gene that promotes tumorigenesis. However, the clinical relevance and potential mechanisms of *CDCA7* in glioma are unclear. *CDCA7* expression level data were obtained from the Chinese Glioma Genome Atlas (CGGA) and The Cancer Genome Atlas (TCGA) databases, and the enriched genes and related signaling pathways were explored. Data on genes in *CDCA7*-related signaling pathways and nine marker genes of ferroptosis were retrieved and a protein–protein interaction (PPI) network analysis was performed. The correlation of *CDCA7* to ferroptosis and tumor infiltration of 22 kinds of human immune cells and the association between *CDCA7* and immune checkpoint molecules were analyzed. *CDCA7* was significantly increased in gliomas in comparison to healthy tissues. Gene Ontology (GO) and gene set enrichment analysis (GSEA) revealed the impact of *CDCA7* expression on multiple biological processes and signaling pathways. *CDCA7* may affect ferroptosis by interacting with genes in the cell cycle pathway and P53 pathway. The increase in *CDCA7* was positively correlated with multiple ferroptosis suppressor genes and genes involved in tumor-infiltrating immune cells and immune checkpoint molecules in glioma. *CDCA7* can be a new prognostic factor for glioma, which is closely related to ferroptosis, tumor immune cell infiltration, and immune checkpoint.

## 1. Introduction

Glioma is the most common intracranial malignant primary tumor. According to the World Health Organization (WHO) glioma classification criteria released in 2007, gliomas are divided into two categories: low-grade gliomas (LGG) (I-II) and high-grade gliomas (HGG) (III-IV) [1,2]. As one of the most challenging cancers in terms of treatment, the efficacy of many therapies, such as surgical resection, adjuvant therapy, molecular targeted therapy, and immunotherapy, is very limited in treating glioma, and its five-year overall survival rate is less than 35% [3,4]. In 2016, for the first time, the WHO classification of Central Nervous System (CNS) tumors used molecular parameters other than histology to define the tumor entity. Isocitrate dehydrogenase (IDH) mutations and 1p/19q combined deletion were considered as the two most critical molecular pathological biomarkers for glioma, and are now widely used in clinical diagnosis and treatment [5]. In 2021, the WHO released the fifth edition of the “Classification of Central Nervous System Tumors” based on biological tumor classification, further adding molecular features to diagnosis and promoting the role of molecular diagnosis in the classification of central nervous system tumors. The classification in this revision allows the diagnosis of glioblastoma in IDH wild-type astrocytoma [6]. Based on grade II or III IDH mutant diffuse astrocytoma, in the WHO classification, revised in 2016, if *CDKN2A/B* homology loss is detected, it will be diagnosed as grade IV IDH mutant astrocytoma. Even without high-level histopathological features, GBM can be diagnosed when at least one of the following molecular features is present: TERTp mutation, EGFR amplification, or the simultaneous presence of +7/−10 [6]. However, due to the molecular heterogeneity of glioma, these biomarkers still have some limitations; thus, the identification of new prognostic markers and therapeutic targets is important to determine cancer risk and improve patient cure rates [5,6,7,8].

Cell division cycle-associated 7 (*CDCA7*), also known as JPO1, is a c-MYC-responsive gene located on chromosome 2q31 [9]. As an MYC target gene, *JPO1*/*CDCA7* is also frequently over-expressed in human cancers [10], but its expression in gliomas has not been studied. Moreover, *JPO1*/*CDCA7* is a direct target of the transcription factor E2F1 [11]. Both MYC and E2F1 are involved in regulating the occurrence and progression of ferroptosis [12,13,14], a newly discovered non-apoptotic cell death pattern characterized by an overwhelming iron-dependent accumulation of lethal lipid reactive oxygen species (ROS) [15]. While the dysregulation of ferroptosis is closely related to tumorigenesis and treatment, according to many previous studies, induction of ferroptosis was found to be effective in treating glioma [16,17,18]. Some studies also indicate that the over-expression of MYC and E2F1 has an inhibitory role in the occurrence of ferroptosis. Given the roles of MYC and E2F1 in regulating the production of *CDCA7*, it is reasonable to assume that *CDCA7* may similarly prevent the ferroptosis of cancer cells. However, this speculation is challenged by the findings of Li et al. and Kim et al., showing that *CDCA7* may facilitate ferroptosis by eliciting activation of the transforming growth factor β (TGF-β) signaling pathway [19,20]. Thus, the role of *CDCA7* in ferroptosis may be in a complex regulatory network that can be affected by many environmental factors. To date, no study is available to elucidate the role of *CDCA7* in ferroptosis and its effect on the occurrence and progression of gliomas.

The role of *CDCA7* in many cancers, such as clear cell renal cell carcinoma (ccRCC), esophageal squamous cell carcinoma (ESCC), colorectal cancer (CRC), and triple-negative breast cancer (TNBC), has been revealed in several previous reports [21,22,23,24]. However, so far, there is no detailed study on the role of *CDCA7* in glioma. In this study, we conducted a comprehensive and systematic bioinformatics analysis of RNA sequencing data retrieved from the Chinese Glioma Genome Atlas (CGGA) and The Cancer Genome Atlas (TCGA) databases and found that the expression of *CDCA7* in gliomas was much higher than that in normal tissues. Meanwhile, the high expression of *CDCA7* was closely related to several clinicopathological parameters of glioma patients. Our analysis further showed that patients with high expression of *CDCA7* generally have shorter survival times. Furthermore, Gene Ontology (GO), gene set enrichment analysis (GSEA), and Kyoto Encyclopedia of Genes and Genomes (KEGG) analysis revealed the biological function of *CDCA7* in glioma and its related enriched genes and pathways. Subsequent protein–protein interaction (PPI) network analysis with nine marker genes of ferroptosis disclosed possible targets or regulators of *CDCA7* that might regulate ferroptosis. In addition, we found that *CDCA7* level has a strong correlation with tumor-infiltrating immune cells and immune checkpoint molecules, revealing its potential function in regulating the prominent biological processes in tumor progression.

## 2. Materials and Methods

### 2.1. Data Collection and Bioinformatics Analysis

The CGGA database (http://www.cgga.org.cn/index.jsp, accessed on 6 January 2023) contains functional genomics data from over 2000 glioma samples of Chinese patients. The database aims to comprehensively map the glioma genome of the Chinese population through genomics technology combined with bioinformatics analysis, explore the drivers and key signaling pathways in the development of glioma, guide molecular typing and drug target development of glioma, and lay the foundation for the development of the whole chain of precision medicine for glioma. We retrieved the mRNAseq_693 (platform form: Illumina HiSeq) and mRNAseq_325 (platform form: Illumina HiSeq 2000 or 2500) datasets from the CGGA database and collected a total of 1018 glioma samples. The “limma” package was used to normalize and batch process both mRNAs of the two platforms. The “survminer” package of R software (version 4.2.2) for survival analysis and the “ggpubr” package for gene expression analysis were used.

TCGA (https://portal.gdc.cancer.gov, accessed on 5 January 2023) is a cancer research project established by the National Cancer Institute (NCI) in collaboration with the National Human Genome Research Institute (NHGRI). It improves scientific understanding of the molecular basis of cancer development and our ability to diagnose, treat, and prevent cancer. Over the past decade, the database has included a wide range of data on more than 20,000 samples from 33 cancers, including transcriptome expression data, genomic variation data, methylation data, clinical data, and more. Sequencing data and clinical data from LGG and glioblastoma multiforme (GBM) were downloaded from the TCGA database and integrated using the “limma” package. Subsequently, we downloaded GTEx normal brain tissue gene expression data from UCSC Xena (https://xena.ucsc.edu/, accessed on 6 February 2023). Based on the GTEx data, we compared the expression of *CDCA7* in different normal tissues using the R language “ggpubr” package. The expression of *CDCA7* was analyzed in 704 glioma samples, 5 normal brain tissue samples from TCGA, and 1152 normal brain tissue samples from GTEx.

### 2.2. Glioma Tissue Microarray, Specimen, and Immunohistochemistry

The glioma tissue microarrays containing 57 samples from glioma patients and 3 normal brain tissue samples were purchased from Shanghai Outdo Biotech Co., Ltd. (Shanghai, China). In addition, 8 normal tissue samples and 17 glioma samples were collected from glioma patients enrolled at the Affiliated Hospital of Nantong University. All procedures in this study complied with the Declaration of Helsinki and were approved by the ethics committees of Nantong University and the Affiliated Hospital of Nantong University. Written informed consent was obtained from all participants involved in this study. All glioma patients were graded according to WHO glioma classification criteria released in 2021. For IHC staining, the sample sections were deparaffinized and hydrated. After blocking with 5% bovine serum albumin (BSA), the samples were then incubated with anti-*CDCA7* primary antibody (goat polyclonal, 1:1000; 15249-1-AP; Proteintech, Wuhan, China) overnight at 4 °C. Subsequently, the sections were incubated with horseradish peroxidase (HRP)-conjugated sheep anti-rabbit IgG at 37 °C for 30 min. Immunoreactivity was developed using diaminobenzidine (DAB) (DAKO, Copenhagen, Denmark) reagents. Finally, the sections were counterstained with hematoxylin. The immunostaining of each specimen was analyzed by two independent pathologists who were blinded to the clinicopathological data of the patients. The pathological scores were determined by the strength and percentage of positive cells. The staining intensity was evaluated as 0 (no staining), 1 (weak staining), 2 (moderate staining), or 3 (strong staining).

### 2.3. Enrichment Analysis

Pearson correlation analysis was conducted in the CGGA and TCGA databases to select the characteristic gene list of genes or cell clusters with a cor > 0.5 and a *p* < 0.05 for *CDCA7* and upload it to the Database for Annotation, Visualization, and Integrated Discovery (DAVID, https://david.ncifcrf.gov, ccessed on 7 February 2023.) to obtain GO analysis results. The first six results in ascending order of *p*-value were used to complete this study. GSEA is a computational method to reveal genomic expression data through basic knowledge [25]. According to the expression level of *CDCA7*, it was divided into a high expression group and a low expression group for enrichment analysis. A total of 1000 genome exchanges were conducted to screen biological pathways related to *CDCA7*. A normalized *p* value < 0.05 and an FDR q value < 0.05 were considered statistically significant.

### 2.4. Screening for Ferroptosis-Associated Genes

The FerrDb database (http://www.zhounan.org/ferrdb, ccessed on 4 March 2023) is now the first manually sorted iron death database, jointly built by the Brain Hospital affiliated with Guangzhou Medical University, the School of Life Sciences of Sichuan University, and the key experimental research team of the Ministry of Biological Resources and Ecological Environment, which is used to manage and identify the markers and regulatory factors related to ferroptosis and ferroptosis-related diseases [26]. We downloaded 9 ferroptosis marker genes and 19 ferroptosis suppressor genes with a score of 3 or greater from the database. Due to the lack of “FTMT” genes in the CGGA database, we finally selected the 18 genes most associated with the ferroptosis suppressor gene score and analyzed them logarithmically.

### 2.5. Protein–Protein Interaction (PPI) Network Analysis

Pathway-related genes of GSEA and the nine marker genes of ferroptosis in the FerrDb database were selected for PPI network analysis. The protein interactions between pathway-related genes and ferroptosis marker genes were analyzed using the STRING online database (https://string-db.org/, ccessed on 20 March 2023).

### 2.6. Statistical Analysis

Statistical analysis was conducted using spss26.0, GraphPad Prism 9.0, and R 4.2.2. The chi-square test was used to analyze the relationship between the expression of *CDCA7* and clinical pathological characteristics. Survival analysis was performed using the logarithmic rank test and Kaplan–Meier method. All statistical tests were bilateral tests, and *p* < 0.05 was statistically significant.

## 3. Results

### 3.1. CDCA7 Shows High Expression in Gliomas and Low Expression in Normal Human Brain Tissues

To investigate the differential expression of *CDCA7* in normal brain tissues and gliomas, we retrieved glioma RNA sequencing datasets from the TCGA and CGGA databases and obtained the expression of *CDCA7* in normal tissues from the GTEx database. Compared with normal tissues, *CDCA7* expression was significantly up-regulated in glioma tissues after analysis of data obtained from both the TCGA (Figure 1A) and CGGA (Figure 1B) databases. Based on data from the GTEx database, we analyzed the expression of *CDCA7* in different normal tissues. The results showed that the abundance of *CDCA7* in normal brain tissues is much lower than that in most other tissues (Figure 1C). The analysis of gender differences in *CDCA7* expression revealed that *CDCA7* levels in male and female brains do not differ significantly (Figure 1D).

For further verification, we conducted immunohistochemical experiments on 11 normal brain tissues, 12 LGG tissues, and 62 HGG tissues. We found that gliomas exhibited higher expression of CDCA7 than normal tissues (Figure 2). Moreover, with the increase in glioma grading, the expression of CDCA7 also increased (Figure 2). Therefore, these findings suggested that *CDCA7* is highly expressed in gliomas, and the increase in its expression is positively correlated with disease grading.

### 3.2. The Relationship between CDCA7 Expression and Clinicopathological Features

The data for the expression levels of *CDCA7* displayed a high individual variation among patients, which might correlate with different clinicopathological symptoms. To clarify if the expression level of *CDCA7* had an impact on patients’ clinical features, we subsequently collected data on clinical features, including 1p/19q codeletion status, IDH mutation status, age, gender, WHO grade, and histological diagnosis of the patients, and found that with the increase in the *CDCA7* expression level, the clinicopathological changes and survival status of patients with glioma showed a trend of deterioration in both CGGA and TCGA databases (Figure 3A,B). Based on these observations, we further analyzed the correlation between *CDCA7* level and the clinical features. We found that the increase in *CDCA7* level positively correlated with a rise in the diagnostic grade of gliomas (Figure 3C,F). Subsequently, we compared the expression levels of *CDCA7* in gliomas bearing a wild-type or mutant IDH and found that *CDCA7* was highly enriched in gliomas carrying a wild-type IDH (Figure 3D,G). Finally, our analysis revealed that *CDCA7* expression levels in gliomas without 1p/19q coding were higher than those in 1p/19q coding samples (Figure 3E,H). Taken together, these findings demonstrated that increased *CDCA7* expression is significantly associated with the increased malignancy of glioma.

### 3.3. The Relationship between CDCA7 Expression and the Prognosis of Glioma Patients

The increased expression of *CDCA7* in glioma promoted our analysis of the relationship between *CDCA7* levels and the survival of glioma patients. Combining RNA sequencing data from the GTEx database, TCGA database, and CGGA database, the data include 1152 normal brain tissues in the GTEx database, 5 normal brain tissues and 704 glioma tissues in the TCGA database, and 1018 glioma tissues in the CGGA database. According to the expression level of *CDCA7*, glioma patients were divided into low-expression and high-expression groups. We found that *CDCA7* levels were negatively correlated with patient survival time. Specifically, analysis of data from the TCGA database showed that the higher expression level of *CDCA7* indicates a lower survival rate and poorer prognosis in glioma patients (Figure 4A). Correspondingly, using data from the CGGA database, we obtained the same conclusion regarding the relationship between *CDCA7* levels and patient survival and prognosis (Figure 4B). In addition, the receiver operating characteristic (ROC) curve was used to analyze the diagnostic value of the *CDCA7* gene in gliomas and revealed that within 1, 3, and 5 years, the *CDCA7* area under the curve (AUC) (Figure 4C) was 0.693, 0.732, and 0.702, respectively, using data from the TCGA datasets, while in the CGGA dataset, the *CDCA7* AUC (Figure 4D) was 0.610, 0.676, and 0.703, respectively, indicating that the expression level of *CDCA7* has good diagnostic value for gliomas.

### 3.4. Enrichment Analysis of CDCA7

The above studies indicated that *CDCA7* could be used as a prognostic factor for glioma, while the mechanisms of action were unclear. To this end, we explored the biological functions of *CDCA7* and screened the genes most related to *CDCA7* in the CGGA and TCGA databases using Pearson correlation analysis (cor > 0.5, *p* < 0.05) and found that the most relevant biological processes to *CDCA7* are cell mitosis, DNA replication, and DNA repair (Figure 5A,B). In addition, the most important cell components are the nucleus, nucleoplast, kinetochore, chromosomes, and centromeric region (Figure 5C,D). The molecular functions are ATP binding, protein binding, DNA binding, and chromatin binding (Figure 5E,F).

To further investigate the mechanisms underlying the role of *CDCA7* in glioma, we conducted a GSEA study to find the possible mechanisms and signaling pathways associated with the enhancement of *CDCA7* in glioma. Several pathways exhibiting significant differential enrichment in gliomas correlated with high *CDCA7* expression phenotypes were identified using the data from the CGGA and TCGA databases based on NES and FDR q-values (FDR < 0.05), respectively (Table 1 and Table 2; Figure 5G,H). Using data from the CGGA database, we identified five pathways associated with highly expressed *CDCA7*, including the nucleotide excision repair pathway, cell cycle pathway, P53 signaling pathway, spliceosome pathway, and homologous recombination pathway. These findings were largely in agreement with the results of analysis using data from the TCGA database, showing that the increased expression of *CDCA7* in gliomas was associated with the enrichment of four pathways, including the cell cycle pathway, nucleotide excision repair pathway, P53 signaling pathway, and homologous recombination pathway, while analysis of the TCGA database also revealed enrichment of the pyrimidine metabolism pathway in gliomas bearing a high expression of *CDCA7*. Together, the above enrichment reveals the biological function of *CDCA7* and its potential role in regulating the activity of these pathways to regulate the progression of glioma.

### 3.5. The Relationship between CDCA7 and Ferroptosis Suppressor Genes in Glioma

Previous studies have reported that ferroptosis is reduced in tissues of glioma patients, and induction of ferroptosis is an effective strategy for treating glioma [16,17,18]. To further verify the role of ferroptosis in glioma, we retrieved the gene expression data of 19 ferroptosis suppressor genes with a confidence score of 3 or greater from the FerrDb website. Due to the lack of the “FTMT” gene in the CGGA database, we analyzed the expression of 18 ferroptosis suppressor genes in gliomas. The analysis was conducted in combination with the GTEx database and CGGA and TCGA databases. We found that most of the ferroptosis suppressor genes in normal and glioma tissues showed significant differences (Figure 6A and Figure 7A). These results indicated that ferroptosis was reduced in gliomas. Then, based on the CGGA database and TCGA database with normal tissue removed, we explored the relationship between *CDCA7* and these differential genes and found that several genes, including NFE2L2, HMOX1, HSPA5, SLC40A1, and STAT3, had positive correlations with *CDCA7* based on data from both the CGGA and TCGA databases (Figure 6B–I and Figure 7B–G). These findings might reveal the possible role of *CDCA7* in regulating ferroptosis and thereby affecting the progression of glioma.

### 3.6. Analysis of Potential Functional Links between CDCA7 and Ferroptosis through PPI

The above GSEA study indicated the possible role of *CDCA7* in regulating several signaling pathways, which may contribute to the progression of glioma. Considering the critical role of ferroptosis in the development of glioma and the undefined role of *CDCA7* in regulating ferroptosis, as mentioned above, we reasoned that *CDCA7* may regulate ferroptosis via these signaling pathways. To verify this speculation, we retrieved the related genes of the cell cycle pathway, nucleotide excision repair pathway, P53 signaling pathway, and homologous recombination pathway that were enriched with *CDCA7* in glioma from GSEA. Meanwhile, we selected nine ferroptosis marker genes (FTH1, GPX4, CHAC1, HSPB1, NFE2L2, PTGS2, SLC40A1, TF, TFRC) from the FerrDb website. Using the STRING database, we explored the potential functional relationship between *CDCA7* and ferroptosis. Genes that can directly interact with both pathway-related genes and ferroptosis marker genes may act as partners of *CDCA7* in regulating ferroptosis. We found that, in the cell cycle pathway, *CDCA7* might regulate ferroptosis by directly interacting with MYC, cyclin-dependent kinase 2 (CDK2), and SFN (Figure 8A), while several genes in the P53 signaling pathway, including RRM2, CDK2, and SFN, might interact with *CDCA7* and thereby regulate the occurrence of ferroptosis (Figure 8B). In contrast, genes of other signaling pathways did not show interactions with ferroptosis. These findings revealed that *CDCA7* may have a role in regulating ferroptosis via interacting with genes in the cell cycle and P53 signaling pathways.

### 3.7. Correlation between CDCA7 Expression Level and Tumor Immune Cell Infiltration

To further explore the potential effect of *CDCA7* on the infiltration of 22 kinds of tumor immune cells in gliomas, we divided the samples into two groups (high and low) based on expression of *CDCA7*. Analysis of data from the CGGA database revealed that *CDCA7* level was positively correlated with the proportions of CD8+ T cells, follicular T helper cells, T cells γ delta, monocytes, macrophages M0, and macrophages M2 (Figure 9A). In contrast, analysis of data from the TCGA database disclosed a correlation between *CDCA7* level and several tumor-infiltrating cell types in gliomas, including T cells CD4 naive, T cells follicular helper, T cells γ delta, monocytes, macrophages M0, macrophages M1, macrophages M2, and dendritic cells (Figure 9C). Subsequently, we performed immune cell correlation analysis on the CGGA database and TCGA database and found that *CDCA7* showed a strong correlation with the infiltration of macrophages M0, T cells γ delta, T cells follicular helper, and monocytes immune cells in both databases (Figure 9B,D).

### 3.8. Association of CDCA7 with Immune Checkpoint Molecules

To further study the relationship between *CDCA7* and the immune microenvironment of glioma tissues, we conducted a more accurate analysis and revealed the relationship between *CDCA7* and the immune checkpoint molecules in glioma. Through analysis of glioma samples in the CGGA and TCGA databases, we found that *CDCA7* expression level was significantly correlated with tumor-related immunosuppressive molecules, including CD80, CD276, CD28, PDCD1LG2, and TNFSF4 (Figure 9E,F). These data suggest that *CDCA7* may act as an immunosuppressive gene and may be a new immunotherapeutic target for glioma.

## 4. Discussion

Glioma, originating from the neuroectoderm, is the most common primary tumor type affecting the human central nervous system [27]. As a major malignant primary brain tumor in adults, the effective management of glioma is one of the most challenging issues clinically, and its total 5-year survival rate is less than 35% [4,28]. In recent years, with the continuous progression of research on the prognostic molecules and therapeutic targets of glioma, targeted therapy and immunotherapy for glioma have become the new research focus [29,30]. Currently, isocitrate dehydrogenase (IDH) levels have prognostic value in wild-type IDH gliomas, and 1p/19q gene expression is a potential prognostic marker in 1p/19q non-deletion gliomas [5]. IDH mutation and 1p/19q codeletion are early genetic events in the progression of glioma [31,32]. However, due to the molecular heterogeneity of glioma, the efficiency of the two markers in predicting the prognosis of glioma is limited; thus, it is necessary to identify new prognostic biomarkers and therapeutic targets [5,7,8].

JPO1/*CDCA7* is a member of the gene family involved in regulating cell division and the cell cycle, encoding a nuclear protein containing 371 amino acids [9,10]. As a responsive gene of c-MYC and E2F1, *CDCA7* participates in tumor transformation [9,11,33]. Similar to MYC, which is often over-expressed in human cancer, *CDCA7* is also over-expressed in several human cancers [10]. In the present study, we analyzed the expression of *CDCA7* in the TCGA, CGGA, and GTEx databases and found that the expression of *CDCA7* in glioma was up-regulated compared with normal tissues. We then studied the relationship between *CDCA7* level and clinicopathologic parameters of glioma patients. In the CGGA and TCGA databases, with the increase in the *CDCA7* expression level, 1p/19q codeletion status, IDH mutation status, age, gender, WHO grade, and histological diagnosis showed a worsening trend. Further analysis showed that the expression of *CDCA7* was positively correlated with tumor grade, wild-type IDH, and 1p/19q non-codeletion in glioma. In addition, we divided glioma patients into two groups based on the expression level of *CDCA7*—a low-expression group and a high-expression group—for survival analysis. In the CGGA and TCGA databases, the OS rate of patients with a high expression of *CDCA7* was markedly lower than that of patients with a low expression of *CDCA7*. These results indicate that a high expression of *CDCA7* indicates a poor prognosis for glioma.

Systemic analysis of the biological functions and signaling pathways of *CDCA7* in gliomas may improve our understanding of the molecular mechanism of this molecule in affecting the occurrence or progression of glioma. Mannino et al. found that glioma stem-cell-like cells are believed to be the cause of treatment resistance and tumor recurrence in glioblastoma patients after chemo-radiation therapy, while glioma stem cells seem to be highly susceptible to abnormal cell division and polyploidization [34]. After analyzing the CGGA and TCGA databases, the GSEA results showed that high expression of *CDCA7* was related to the P53 signaling pathway, cell cycle pathway, nucleotide excision repair pathway, and homologous recombination pathway. P53 is a tumor suppressor gene that has a role in inhibiting DNA replication [35], regulating apoptosis [36], slowing down cell proliferation [37], and controlling cell motility and invasion [38]. An increasing body of evidence shows that the expression of P53 is closely related to the occurrence and development of glioma and is considered to be one of the extensive biomarkers for human glioma [39,40]. P53 can regulate cell cycle arrest and then regulate the proliferation of glioma cells [36,41,42]. At the same time, Dai et al. found that the drug resistance of glioblastoma cells can be enhanced by promoting the repair of nucleotide excision [43]. Targeted homologous recombination pathways can improve the sensitivity of glioma to ionizing radiation and improve the survival rate of patients [44]. These results suggest that *CDCA7* may affect the development of glioma by regulating the activity of multiple signaling pathways.

Ferroptosis is a newly emerging form of programmed cell death characterized by the accumulation of iron-dependent lipid peroxides [15,45]. Many studies have shown that induction of ferroptosis can effectively treat many types of cancers, including glioma [46]. Therefore, ferroptosis may be a new potential target for the treatment of glioma [16,17,18]. According to other reports, *CDCA7* could enhance the activation of TGF-β and thereby promote ferroptosis of hepatocellular carcinoma cells by repressing the expression of SLC7A11 and enhancing lipid peroxidation [19,20], which is inconsistent with our inference that *CDCA7* inhibits ferroptosis in gliomas. Thus, the role of *CDCA7* in ferroptosis may be dependent on its tumor microenvironment, which integrates various regulatory factors and thus determines the fate of the cells. However, direct evidence that aims to study the role of *CDCA7* in ferroptosis is still absent. We obtained 18 genes most related to ferroptosis suppressors from the FerrDb website, with confidence scores greater than or equal to 3, to explore the relationship between *CDCA7* and ferroptosis. By analyzing the data of GTEx combined with the CGGA and TCGA databases, it is evident that almost all ferroptosis suppressor genes show differential expression between normal tissues and glioma samples. Further analysis of these differential genes reveals that the level of several genes, including NFE2L2, HMOX1, HSPA5, SLC40A1, and STAT3, is positively correlated with *CDCA7* level in both databases. This observation suggests that the possible inhibitory effect of excessive *CDCA7* on ferroptosis may be the cause of the poor prognosis of glioma. In addition, as mentioned above, the exact role of *CDCA7* in ferroptosis needs to be further studied since this molecule is involved in pathways that play opposing functions in regulating ferroptosis.

To further validate the role of *CDCA7* in ferroptosis and glioma, we retrieved genes related to the cell cycle pathway and P53 signaling pathway from GSEA and conducted PPI analysis with nine marker genes for ferroptosis. It was found that *CDCA7* may have an impact on ferroptosis through interacting with several genes in the cell cycle pathway, such as MYC, CDK2, and SFN. Moreover, *CDCA7* may interact with several genes in the P53 signaling pathway, such as RRM2, CDK2, and SFN, and thereby regulate the development of ferroptosis. Several earlier studies indicated that the over-expression of MYC could promote the expression of *CDCA7* and thus play a role in suppressing the occurrence of ferroptosis [12,13,14]. In this regard, *CDCA7* may work as a regulator, downstream of MYC, and suppress the ferroptosis of cancer cells. Similarly, studies have shown that CDK2, SFN, and RRM2 can inhibit ferroptosis [14,47,48]. These studies may reveal the targets of *CDCA7* and the association between *CDCA7* and ferroptosis in glioma. In addition, the potential role of *CDCA7* in ferroptosis and its effects on glioma need to be confirmed in both in vitro and in vivo experiments.

The expression of *CDCA7* was significantly correlated with tumor-infiltrating immune cells. In both the CGGA and TCGA databases, cells with a strong correlation with *CDCA7* included macrophages M0, γδT cells, T cells follicular helper, and monocytes immune cells. These findings were in agreement with observations from several previous reports, such as one by Huang et al. showing that M0 macrophages, as a feature of glioblastoma malignant tumors, are highly enriched in malignant gliomas [49]. Another study, by Yue et al., showed that the expression of immunosuppression-related molecules on the surface of γδ1 T cells was significantly increased in glioma, while the expression of killing-function-related molecules and signal pathways in γδ2 T cells was significantly decreased [50]. Moreover, the infiltrated follicular helper T cells play a role in inhibiting the development of regulatory B cells, leading to inflammation and glioma development [51,52]. In addition, the infiltrated monocytes are highly plastic immune cells that regulate anti-tumor immunity via various mechanisms [53]. Therefore, our study shows that *CDCA7* is closely related to the infiltration of multiple immune cells involved in cancer progression, and thus provides further evidence for the prognostic role of glioma.

Additionally, the analysis of the CGGA and TCGA databases also showed that *CDCA7* was closely related to several tumor-related suppressors, including CD80, CD276, CD28, PDCD1LG2, and TNFSF4, which contribute to tumor development via multiple mechanisms. For instance, CD80 signal transduction can delay tumor proliferation and induce apoptosis by up-regulating the expression of pre-apoptotic molecules and down-regulating anti-apoptotic molecules [54]. CD276 is one of the most important immune checkpoints in the CD28 and B7 superfamily, and its expression is closely related to T cell infiltration, immune checkpoint genes, and immune regulatory interactions between lymphocytes and non-lymphocytes [55]. CD276 is strongly expressed in most types of cancer and is associated with the poor prognosis of glioma [55,56]. The ligand of programmed death 1 (PD-1) (CD274 and PDCD1LG2) is the key mechanism leading to tumor immune escape [57,58]. TNFSF4 is closely related to the induction of anti-tumor immunity [59]. The association between *CDCA7* and these tumor immune checkpoint molecules suggests that *CDCA7* can also be used as a valuable biomarker for predicting prognosis and an immunotherapy target for glioma.

However, our research still has several limitations. Firstly, the data obtained from the CGGA and TCGA databases are retrospective, and the associated clinical records of the patients are limited, which may reduce the precision of our study to a certain extent. Secondly, the integrity and homogeneity of the data are sometimes not guaranteed, which may affect the overall quality of our results. Thirdly, some studies have shown that *CDCA7* can promote ferroptosis, contrary to the function of the MYC pathway in inhibiting ferroptosis. The complexity of the *CDCA7* function in ferroptosis is worthy of further study. Finally, the molecular mechanism by which *CDCA7* regulates the progression of glioma is unclear. Does it play a role internally (related to tumor proliferation, migration, and invasion), externally (immunosuppression), or both? Further studies are required to clarify this issue in both in vitro and in vivo investigations.

## 5. Conclusions

In conclusion, our study revealed an elevated expression of *CDCA7* in gliomas and investigated the biological functions of *CDCA7*. The expression of *CDCA7* is a significant indicator of the poor prognosis of glioma patients. The expression level of *CDCA7* in glioma is positively correlated to disease grade, IDH mutation status, and 1p/19q codeletion. The P53 signaling pathway, cell cycle pathway, nucleotide excision repair pathway, and homologous recombination pathway may be the main pathways of *CDCA7* in regulating glioma. In addition, we also revealed the potential role of *CDCA7* in regulating ferroptosis by interacting with several ferroptosis suppressor genes in gliomas and identified potential targets of *CDCA7* in gliomas through PPI analysis. Further research also found that the expression level of *CDCA7* correlates with tumor infiltration of immune cells and the expression of immune checkpoint molecules, which may indicate a role for *CDCA7* as a new immunotherapeutic target in treating gliomas. Therefore, *CDCA7* may be a promising prognostic biomarker and a target for the clinical treatment of glioma.

## Figures and Tables

**Figure 1 genes-14-01406-f001:**
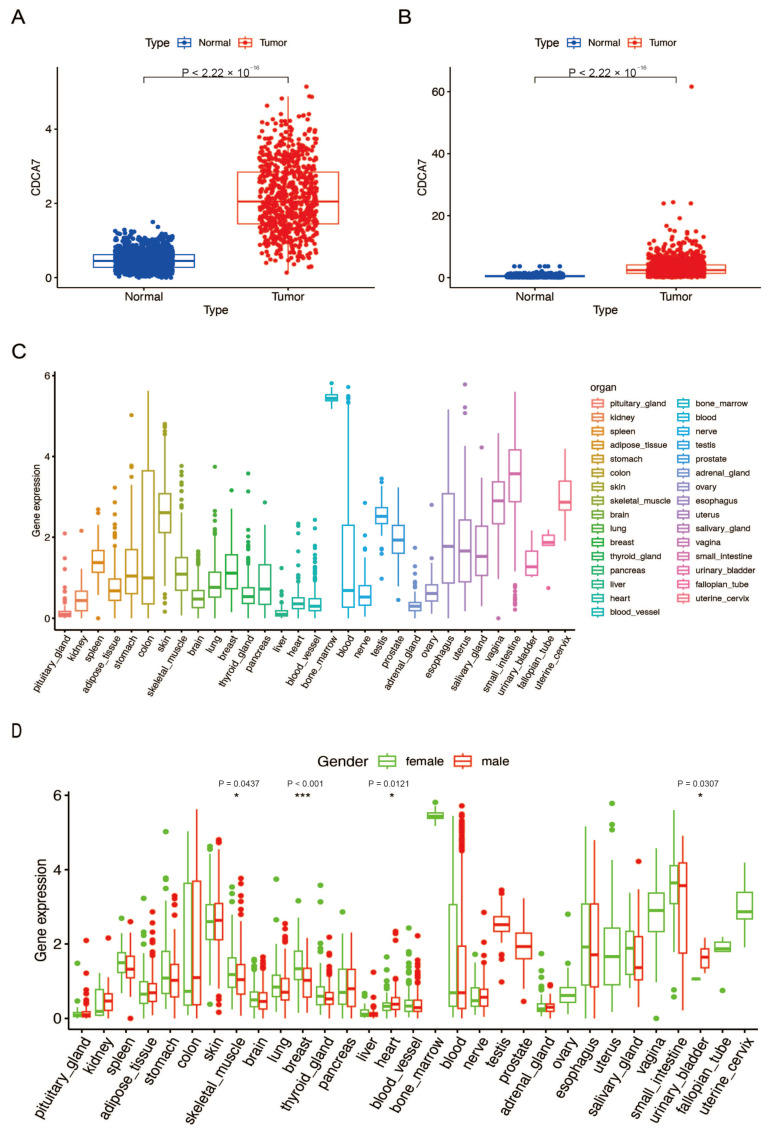
*CDCA7* was over-expressed in gliomas and low-expressed in normal tissues. (**A**) Differences in *CDCA7* expression between glioma tissues (low-grade gliomas (LGG) + glioblastoma multiforme (GBM)) in The Cancer Genome Atlas (TCGA) database and normal tissues in the GTEx database. (**B**) Differences in *CDCA7* expression between glioma tissues in the Chinese Glioma Genome Atlas (CGGA) database and normal tissues in the GTEx database. (**C**) Expression levels of *CDCA7* in various normal human tissues analyzed by GTEx data. (**D**) Expression levels of *CDCA7* in various tissues of different genders. * *p* < 0.05; *** *p* < 0.001.

**Figure 2 genes-14-01406-f002:**
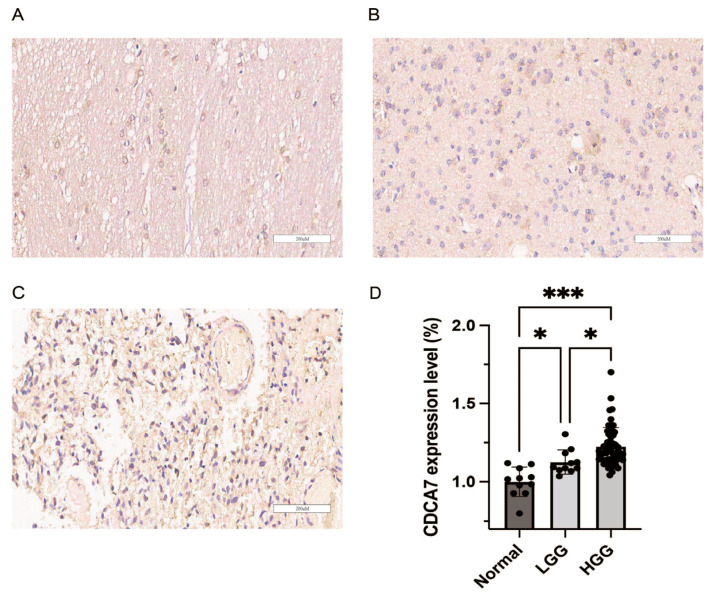
The protein level of CDCA7 was associated with the disease grading of glioma patients. Tissue microarray was used to detect the expression levels of CDCA7 in the normal cerebral cortex (**A**), LGG (**B**), and HGG (**C**). (**D**) Differential analysis of CDCA7 expression in normal tissues, LGG, and HGG (Normal: n = 11; LGG: n = 12; HGG: n = 62). * *p* < 0.05; *** *p* < 0.001, *p* (Normal vs. LGG) = 0.0305, *p* (Normal vs. HGG) < 0.001, *p* (LGG vs. HGG) = 0.0261.

**Figure 3 genes-14-01406-f003:**
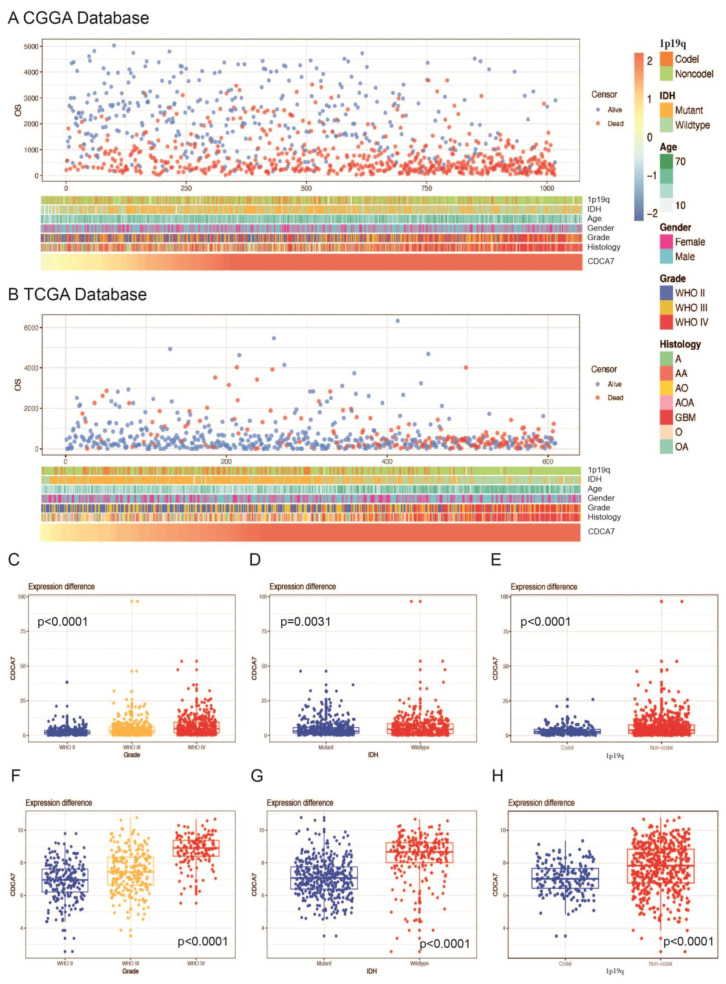
Relationship between *CDCA7* expression and clinicopathological features of glioma. (**A**) Profile of *CDCA7* and clinicopathological features of glioma in the CGGA database. (**B**) Profile of *CDCA7* and clinicopathological features of glioma in the TCGA database. (**C**) Grade in CGGA. (**D**) IDH mutation status in CGGA. (**E**) 1p/19q codeletion status in CGGA. (**F**) Grade in TCGA. (**G**) IDH mutation status in TCGA. (**H**) 1p/19q codeletion status in TCGA.

**Figure 4 genes-14-01406-f004:**
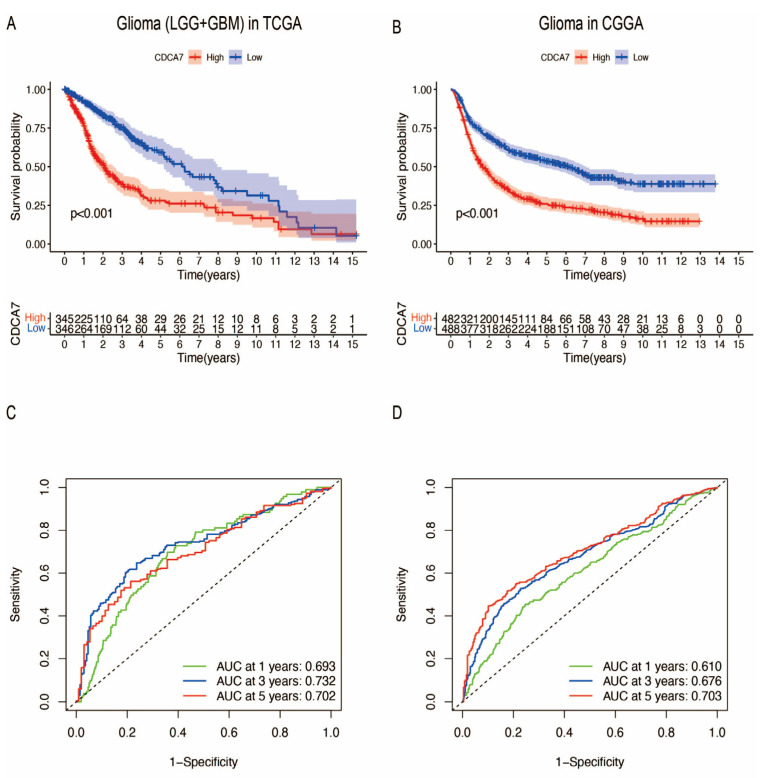
Survival and prognostic analysis of *CDCA7* in gliomas. (**A**) Kaplan–Meier curves of overall survival in the TCGA database. (**B**) Kaplan–Meier curves of overall survival in the CGGA database. (**C**) Receiver operating characteristic (ROC) analyses revealed the predictive value of *CDCA7* in glioma based on the TCGA database. (**D**) ROC analyses revealed the predictive value of *CDCA7* in glioma based on the CGGA database.

**Figure 5 genes-14-01406-f005:**
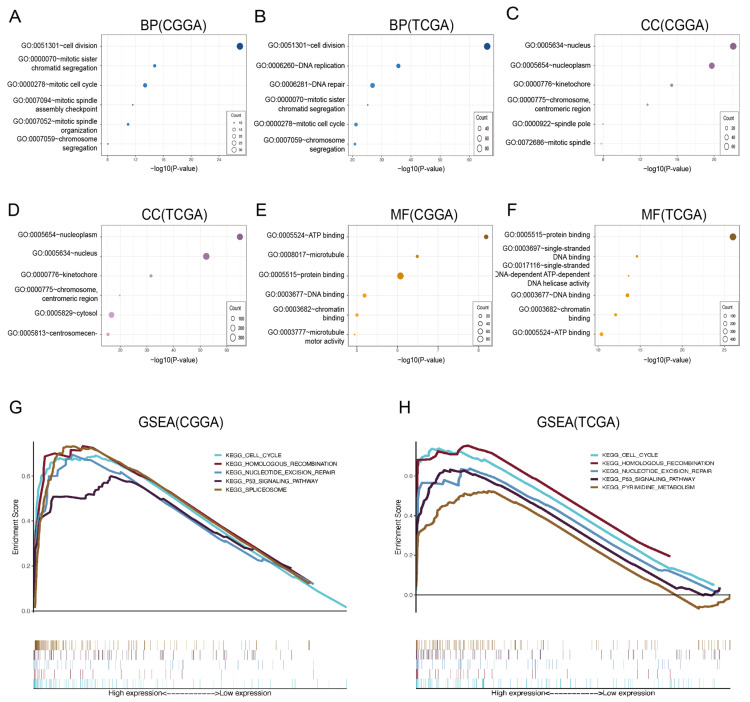
Enrichment plots of Gene Ontology (GO) and gene set enrichment analysis (GSEA). (**A**,**B**) The relationship between biological processes (BP) and *CDCA7* in the CGGA and TCGA databases. (**C**,**D**) The relationship between cellular components (CC) and *CDCA7* in the CGGA and TCGA databases. (**E**,**F**) The relationship between molecular functions (MF) and *CDCA7* in the CGGA and TCGA databases. (**G**) GSEA of the CGGA database. (**H**) GSEA of the TCGA database.

**Figure 6 genes-14-01406-f006:**
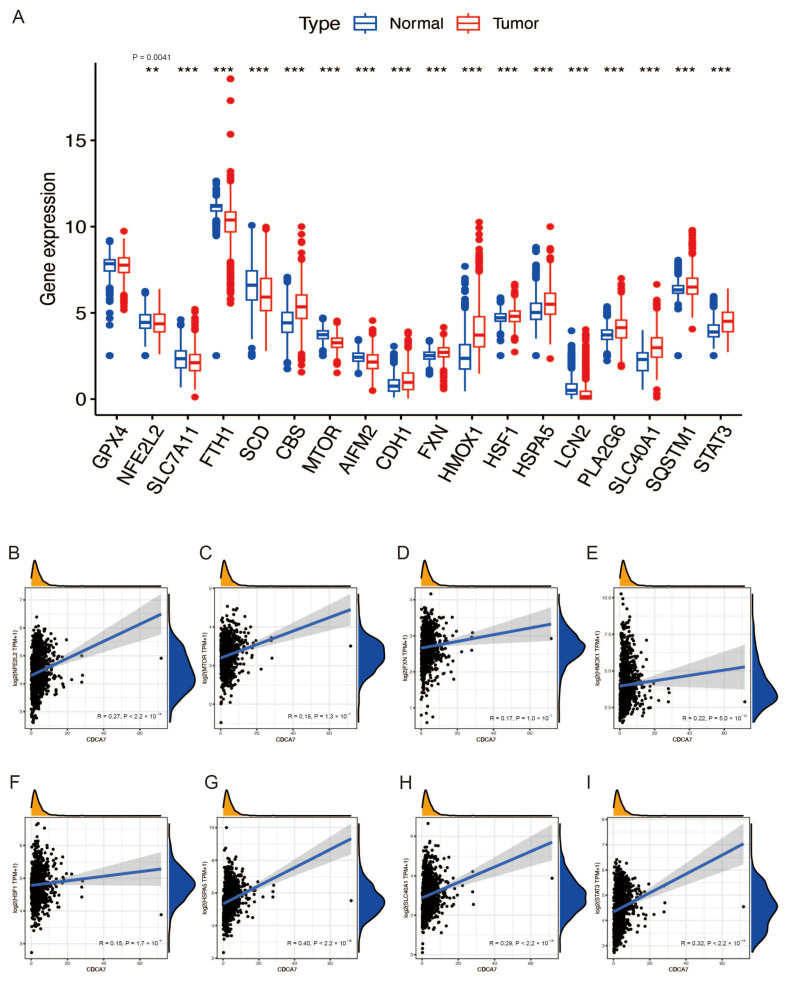
Expression of the ferroptosis suppressor gene in glioma and its relationship with *CDCA7* in the CGGA database. (**A**) Expression of the ferroptosis suppressor gene in normal and glioma tissues in CGGA combined with the GTEx database. (**B**–**I**) Association of *CDCA7* and ferroptosis differentially expressed genes in the CGGA database. ** *p* < 0.01; *** *p* < 0.001.

**Figure 7 genes-14-01406-f007:**
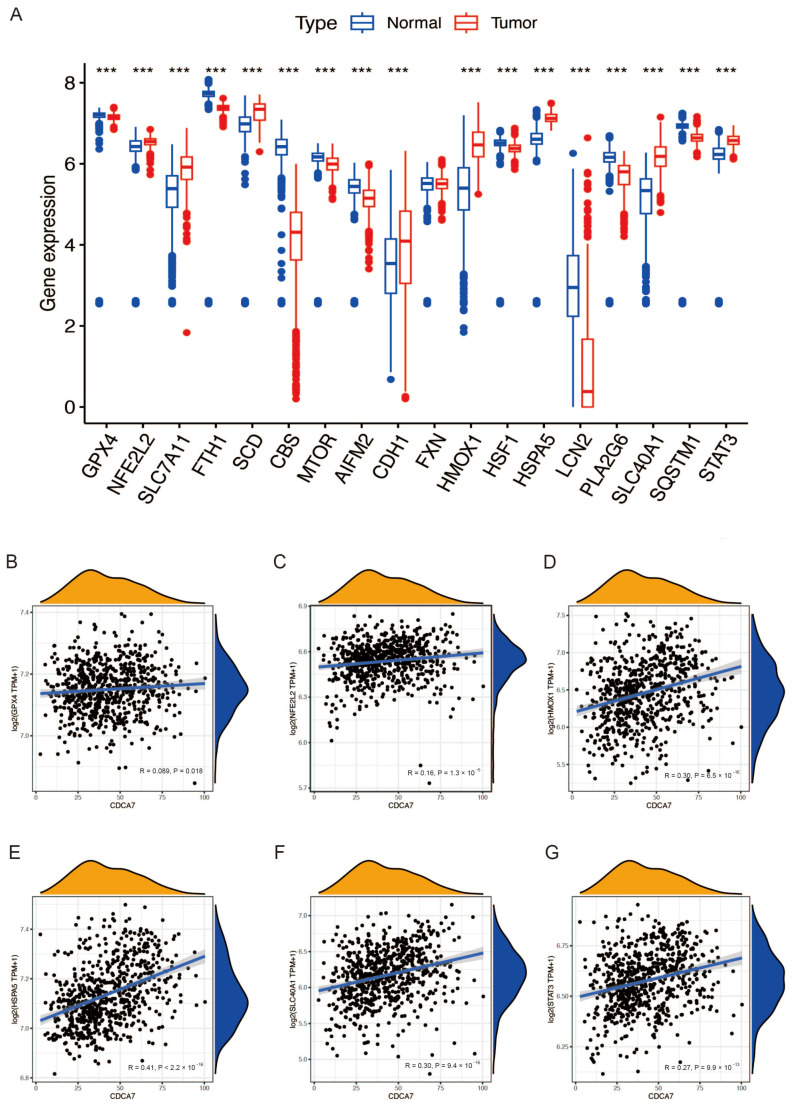
Expression of the ferroptosis suppressor gene in glioma and its relationship with *CDCA7* in the TCGA database. (**A**) Expression of the ferroptosis suppressor gene in normal and glioma tissues in TCGA combined with the GTEx database. (**B**–**G**) Association of *CDCA7* and ferroptosis differentially expressed genes in the TCGA database. *** *p* < 0.001.

**Figure 8 genes-14-01406-f008:**
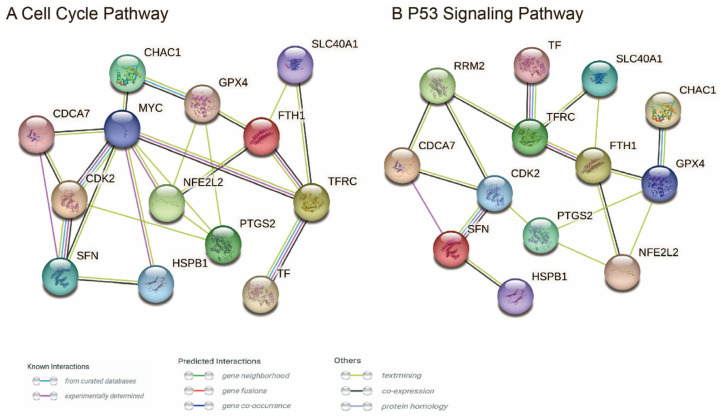
Interaction between *CDCA7* pathway-related genes and ferroptosis marker genes. (**A**) PPI between *CDCA7* and ferroptosis in the cell cycle pathway. (**B**) PPI between *CDCA7* and ferroptosis in the P53 signaling pathway.

**Figure 9 genes-14-01406-f009:**
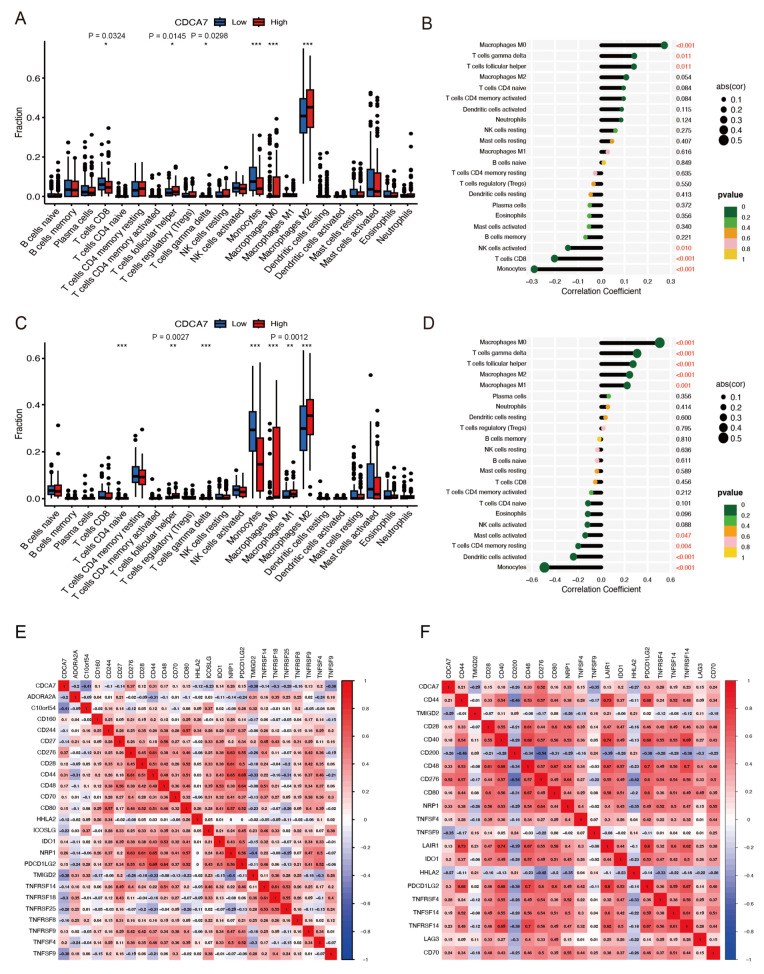
Differential and association analysis of *CDCA7* with tumor-infiltrating immune cells and immune checkpoint analysis in glioma. (**A**) Analysis of *CDCA7* in the CGGA database for tumor-infiltrating immune cell differences in glioma. (**B**) Analysis of *CDCA7* in the CGGA database for tumor-infiltrating immune cell correlation in glioma. (**C**) Analysis of *CDCA7* in the TCGA database for tumor-infiltrating immune cell differences in glioma. (**D**) Analysis of *CDCA7* in the TCGA database for tumor-infiltrating immune cell correlation in glioma. (**E**) The relationship between *CDCA7* and immune checkpoints in the CGGA database. (**F**) The relationship between *CDCA7* and immune checkpoints in the TCGA database. * *p* < 0.05; ** *p* < 0.01; *** *p* < 0.001.

**Table 1 genes-14-01406-t001:** Gene set enrichment analysis (GSEA) of *CDCA7* in glioma in CGGA.

Gene Set Name	NES	Nominal *p*-Value	FDR *q*-Value
KEGG_NUCLEOTIDE_EXCISION_REPAIR	2.27	0.000	0.000
KEGG_CELL_CYCLE	2.21	0.000	0.001
KEGG_P53_SIGNALING_PATHWAY	2.04	0.000	0.003
KEGG_SPLICEOSOME	2.23	0.000	0.000
KEGG_HOMOLOGOUS_RECOMBINATION	2.09	0.000	0.001

**Table 2 genes-14-01406-t002:** Gene set enrichment analysis (GSEA) of *CDCA7* in glioma in TCGA.

Gene Set Name	NES	Nominal *p*-Value	FDR *q*-Value
KEGG_CELL_CYCLE	2.34	0.000	0.000
KEGG_NUCLEOTIDE_EXCISION_REPAIR	2.16	0.000	0.005
KEGG_P53_SIGNALING_PATHWAY	2.13	0.000	0.006
KEGG_HOMOLOGOUS_RECOMBINATION	2.01	0.000	0.014
KEGG_PYRIMIDINE_METABOLISM	2.12	0.000	0.006

## Data Availability

The datasets analyzed during the current study are available in the TCGA database (https://portal.gdc.cancer.gov), CGGA database (http://www.cgga.org.cn/index.jsp), GTEx database (https://xena.ucsc.edu/), DAVID (https://david.ncifcrf.gov), FerrDb database (http://www.zhounan.org/ferrdb), and STRING online database (https://string-db.org/).

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
