# Peer review of "High Expression of CDCA7 in the Prognosis of Glioma and Its Relationship with Ferroptosis and Immunity"

_genes, 2023, doi:10.3390/genes14071406_

Round 1
Reviewer 1 Report
The paper entitled “The high expression of CDCA7 predicts a poor prognosis for glioma and explores its relationship with ferroptosis and immunity” proposed that the CDCA7 can be a new prognostic factor for glioma and it is related to ferroptosis, tumor immune cell infiltration and immune checkpoint. The author did a throughout research about the role of CDCA7 in glioma but some of the experiments can be improved or justified. For example, when they compare the normal samples and the tumor samples, the size of normal samples is way smaller than tumor samples. In the paper, they mentioned that gender also influence the expression pattern of CDCA7 but they didn’t provide any evidence to show the statistic significance.
Also, The way of presenting data is not consistent. In some figures, they provided the specific P values but not in all figures. I would like to suggest the authors to provide all P values.
Overall, this paper provides useful information but I hope the authors could improve these details.
There are some grammar mistakes and some of the descriptions are not accurate. Please fix them
Author Response
We thank the reviewer's comments, please see the attached document for our responses.

Reviewer 2 Report
Thank you for the opportunity to review “The high expression of CDCA7 predicts a poor prognosis for glioma and explores its relationship with ferroptosis and immunity” by Wang, Zhao et al.
In this publication the authors describe analysis of CDCA7 transcript levels in samples from The Cancer Genome Atlas (TCGA) and the Chinese Glioma Genome Atlas (CGGA). The authors identify increased levels of CDC7A in tumour versus normal control tissue, and that increased expression correlates statistically with increased tumour grade and worse survival. Comparison of tumour grade and CDCA7 expression was verified at the protein level by immunohistochemistry and the predictive value of CDCA7 expression for survival was revealed by receiver operating characteristic analysis.
The authors go on to show increased transcript levels of genes involved in the cell cycle, nucleotide excision repair, P53 signalling and homologous recombination based on analysis of gene set enrichment analysis. Based on the literature the authors then explore expression of genes involved in ferroptosis and identify increased transcript levels of genes including NFE2L2, HMOX1, HSPA5, SLC40A1 and STAT3 in samples with high CDC7A expression.
Finally, the authors identify transcript levels of CDC7A correlate with different tumour infiltrating immune cell populations.
The authors provide a clearly argued and statistically robust analysis of the database information available to support their case. However the major weakness of the manuscript ie. the lack of functional confirmation of their findings in vitro and in vivo is identified by the authors in the discussion (lines 447 and 448). Without this the authors have identified correlations rather than mechanisms of action and this limits the interest of the manuscript.
Major Comments
- As the authors appear aware, this manuscript would be greatly strengthened by the incorporation of in vitro and in vivo data. The gene does not appear to be readily amenable to generation of conventional knockout models so use of inducible systems to generate knockouts or mutants may provide a method for this analysis. Even without generation of an animal model, knockout and overexpression of CDC7A in glioma cell lines (again potentially using inducible systems if required) and analysis in culture and/or after implantation in mice would provide some mechanistic information to support their findings. Extending cell line analysis to animal implantation models would also provide a pathway to analyse the importance of different immune cell populations if different strains of immunodeficient host mice were used.
Minor comments are also as follows:
- In the introduction the authors only discuss WHO 2007 and 2016 CNS tumour classifications. They need to discuss the data set in the context of the WHO 2021 tumour classification as changes to high and low grade glioma classification in this update as these will impact their samples.
- In Figure 1D, data for CDCA7 expression is only shown for female bone marrow. Does this reflect a gap in the data available or an error during figure generation?
- Some minor English language editing is required in the abstract and introduction.
Minor English language editing required particularly for abstract and introduction.
Author Response

(The authors gave the same response as above.)

Round 2
Reviewer 2 Report
Thank you for providing an updated manuscript for “The high expression of CDCA7 predicts a poor prognosis for glioma and explores its relationship with ferroptosis and immunity” by Wang et al. Overall the authors have made excellent attempts to address my review questions.
I appreciate that the in vitro and in vivo studies discussed in my review require substantial time, labour and resources and may be beyond the scope of the current study. However I am pleased to hear the authors are planning this work.
I thank the authors for discussing the dataset in the context of the WHO 2021 classification and for confirming that the CDCA7 expression in female bone marrow only reflects a gap in the available data.
The authors may need to update the informed consent and Institutional Review board statement to reflect the consent required for the tissue samples obtained from the Affiliated Hospital of Nantong University.